# Age, Dose, and Locomotion: Decoding Vulnerability to Ketamine in C57BL/6J and BALB/c Mice

**DOI:** 10.3390/biomedicines11071821

**Published:** 2023-06-25

**Authors:** Wen-Chien Chen, Tzong-Shi Wang, Fang-Yu Chang, Po-An Chen, Yi-Chyan Chen

**Affiliations:** 1Department of Psychiatry, Taipei Tzu Chi Hospital, Buddhist Tzu Chi Medical Foundation, New Taipei City 231, Taiwan; 2Department of Psychiatry, China Medical University Hsinchu Hospital, China Medical University, Hsinchu 302, Taiwan; 3Department of Psychiatry, School of Medicine, Tzu Chi University, Hualien 970, Taiwan

**Keywords:** ketamine, NMDA, pharmacological response, locomotion, psychosis, learning, age, strain, dose effect, genetic diversity

## Abstract

Ketamine has been abused as a psychedelic agent and causes diverse neurobehavioral changes. Adolescence is a critical developmental stage but vulnerable to substances and environmental stimuli. Growing evidence shows that ketamine affects glutamatergic neurotransmission, which is important for memory storage, addiction, and psychosis. To explore diverse biological responses, this study was designed to assess ketamine sensitivity in mice of different ages and strains. Male C57BL/6J and BALB/c mice were studied in adolescence and adulthood separately. An open field test assessed motor behavioral changes. After a 30-min baseline habituation, mice were injected with ketamine (0, 25, and 50 mg/kg), and their locomotion was measured for 60 min. Following ketamine injection, the travelled distance and speed significantly increased in C57BL/6J mice between both age groups (*p* < 0.01), but not in BALB/c mice. The pattern of hyperlocomotion showed that mice were delayed at the higher dose (50 mg/kg) compared to the lower dose (25 mg/kg) of ketamine treatment. Ketamine accentuated locomotor activation in adolescent C57BL/6J mice compared to adults, but not in the BALB/c strain. Here, we show that ketamine-induced locomotor behavior is modulated by dose and age. The discrepancy of neurobehaviors in the two strains of mice indicates that sensitivity to ketamine is biologically determined. This study suggests that individual vulnerability to ketamine’s pharmacological responses varies biologically.

## 1. Introduction

Ketamine has been used as a dissociative anesthetic in human medicine since 1970. Growing research demonstrate that it has promising therapeutic effects for pain and treatment-resistant depression [1,2,3]. It can also be used recreationally as a party drug due to its hallucinogenic and dissociative effect for producing different states of mind, emotion, perception, and psychomotor behaviors. The acute effects of ketamine can induce trance-like states such as amnesia, visual distortion, hallucination, dissociative experiences, and psychotomimetic features [1,4]. Chronic abuse can influence the neuropsychological function and cognitive performance and induce urinary toxicity such as cystitis and lower urinary track syndrome [4,5]. An epidemiological survey showed an increasing prevalence of ketamine abuse in Asia, especially for teenagers [6]. Facing the public health and long-term impacts of ketamine abuse is a critical issue. Abuse behaviors not only potentiate the addiction tendency but also bring negative psychiatric consequences to the adolescent [4]. 

Ketamine principally acts as an *N*-methyl-D-aspartate (NMDA) receptor agonist and inhibits glutamatergic neurotransmission, which is responsible for anesthetic, analgesic, and psychotomimetic effects [7]. Moderations of the glutamate neuropathway can produce diverse neurophysiological and neurobehavioral changes, such as impairment of memory encoding and development of psychosis [8]. Growing evidence demonstrates the effectiveness for treatment-resistant depression through NMDA receptor inhibition and downstream α-amino-3-hydroxy-5-methyl-4-isoxazolepropionic acid (AMPA) receptor activation [9,10]. The prefrontal cortex, subgenual anterior cingulate cortex, and hippocampus are thought to be involved in this antidepressant effect [9,11]. Some studies have demonstrated overlapping neuronal circuits underlying depression and addiction, with a specific focus on the mesolimbic projection [10]. Recent studies have shown different pharmacodynamic properties between R- and S-ketamine regarding the risk of adverse effects [9,12]. Nonetheless, the long-term adverse effects of ketamine are worth noting in different populations. Based on the pharmacological properties, activation of the NMDA receptor can enhance long-term potentiation (LTP) and synaptic plasticity, which are important mechanisms for memory consolidation and recall. Compared to receptor activation, the inhibitory effects of ketamine on the NMDA receptor may deter LTP formation and cause negative impacts on memory and learning [13,14]. There are common symptoms of dissociation, hallucination, and delusion in acute ketamine users and cognitive deficits in chronic abusers. Many neuroscience studies have indicated that ketamine-induced psychotomimetic features are strongly correlated with NMDA hypofunction [15,16]. In a clinical observation survey, ketamine exacerbated psychotic symptoms in patients with schizophrenia [17]. Until now, ketamine-induced neurobehavioral change has seemed to be a powerful animal model for schizophrenia research [18,19].

Adolescence is a critical stage of brain development from the childhood to adulthood, possessing characteristics of rapid neuronal differentiation and neurocircuit remodeling [20]. During this stage, the brain is remarkably vulnerable to stressors and diverse substances, which can predispose the adolescent to many neuropsychiatric problems such as schizophrenia and bipolar disorder and may increase risk-taking behaviors, which can easily lead to substance addiction [20,21,22]. The putative mechanism of ketamine exerts negative impacts on information processing and episodic memory formation [23]. Moreover, ketamine abuse is often accompanied with adverse psychotomimetic effects in adolescent. On the other hand, growing evidence has demonstrated the efficacy of ketamine on treatment-resistant depression, which has a higher prevalent rate in adults and the elderly. In recent years, research into different treatments for major depressive disorders have led to a more comprehensive understanding of the mechanisms of depression beyond the monoamine hypothesis. For instance, studies involving ketamine have demonstrated the involvement of the glutamatergic system, while research on brain stimulation has revealed the significance of thalamocortical dysrhythmia [24,25]. Ketamine may prove to be a valuable tool, as it may help us to understand the pathophysiology of depression and identify possible biomarkers, such as genetic or electroencephalography biomarkers [26], in the future. Thus, the trade-off between the risks and benefits of ketamine treatment between the adolescent and the adult with refractory depressive features is a matter of dispute. To clarify, the age effect on the differences of ketamine sensitivity and pharmacological response is an important issue. In another aspect, geographical differences of ketamine abuse may also reflect the pharmacogenetic variation. 

To explore the age effect and strain differences on the ketamine pharmacological response, an open field test was used as an animal model to evaluate neurobehavioral changes after acute ketamine injection. The mouse strains of BALB/c and C57BL/6J were used to compare the ketamine sensitivity between adolescence and adulthood. The study was designed (1) to investigate age-related behavioral changes following ketamine injection; (2) to compare the strain differences in terms of ketamine sensitivity; and (3) to compare the interactions of ketamine dose effects between strains and different age groups.

## 2. Materials and Methods

The study was designed to investigate the locomotor neurobehavioral changes following ketamine injection in mice with different ages and strains. An open field test with a computerized video-tracking system was adapted to measure the locomotor changes [27,28,29]. 

### 2.1. Subjects

Subjects were male C57BL/6J and BALB/c mice obtained from the National Laboratory Animal Center, Taiwan. C57BL/6J was chosen as a reference strain given its common use in models of explorative behavioral studies [27,28]. BALB/c mice, as genetic backgrounds for mutants, were chosen based on their frequent use in behavioral neuroscience, including sensitivity to stress and expectations of different reactions to ketamine [28,29]. Adolescent (4–6 week-old) and adult (9–15 week-old) mice were used. Mice were housed in groups of four in a temperature (22 ± 1 °C)- and humidity (50 ± 5%)-controlled vivarium under a 12-h light/dark cycle with ad libitum access to food and water. All experimental procedures were approved by the Institutional Animal Care and Use Committee, Buddhist Taipei General Hospital, and were conducted following the regulation of reduction and refinement. All efforts were made to minimize the number of animals used and their suffering.

### 2.2. Drug Preparation

Testing doses of ketamine were based on our pilot work [30,31] and previous studies of behavioral effects in mice [32,33,34,35]. In this experiment, the study dose was first assessed by the rotarod motor test to evaluate motor balance. To determine the doses without the effects of anesthesia and paralysis, testing doses were determined to the sub-anesthesia range from 25 to 50 mg/kg, with 0 mg/kg used as the negative control. The use of ketamine was approved by the Taiwan Food and Drug Administration, and ketamine was purchased from Pfizer (New York, NY, USA). Ketalar (ketamine hydrochloride, 50 mg/mL) was dissolved in 0.9% saline vehicle and intraperitoneally injected in volumes of 10 mL/kg. 

### 2.3. Study Procedure

The novel open field apparatus was a 40 × 40 × 35 cm square arena with opaque Plexiglas walls and floor. The white composition was designed for C57BL/6J strain mice and the black for BALB/c mice. Before the behavioral study, mice were transferred to the testing room 1 h prior to testing for acclimation to the test environment. The open field apparatus was wiped with 70% ethanol prior to each trial and between trials. The study procedures (Figure 1) were performed in adolescent and adult stages of both strains with randomized assignment of different ketamine doses.

In the test, the open field apparatus was evenly illuminated to ~50 ± 5 lux [36]. A cross-over design was used for the mice to avoid environmental differences in the open field and to reduce the number of animals used. The apparatus was designed with black and white colors for C57BL/6J and BALB/c mice, respectively, to adjust the color differences for two strains under the monitoring system. The mice were initially placed in the apparatus for a 30-min habituation period. This design served as both an internal and systemic control. After 30 min of baseline locomotion habituation, mice were followed by intraperitoneal injection (i.p.) with ketamine (0, 25, and 50 mg/kg). The mice then were placed in the perimeter and allowed to explore the apparatus for 60 min. A computerized video tracking system (SINGA Trace MouseⅡAnubis Track 1.6.4 pro) was adapted to measure the motor responses following ketamine injection.

### 2.4. Statistical Analysis

All the statistical analyses were conducted using GraphPad Prism (GraphPad Software, Inc., San Diego, CA, USA) version 5.01.336 and Statistical Package for Social Science version 18 (SPSS Inc., Chicago, IL, USA). The moving paths and travelled distances in the open field were tracked before and after ketamine injection. The measurement parameters were analyzed by analysis of variance (ANOVA), followed by Bonferroni’s post hoc tests. Probability (*p*) values under 0.05 were considered significant.

## 3. Results

Two strains of mice during adolescence and adulthood were randomized to different ketamine doses (0, 25, 50 mg/kg, separately) by i.p. injection. The number of experimental mice in each group was 11–12 (by strain × age × ketamine dose). Prior to the drug treatment, there were no differences in the baseline travelled distance within each strain × age group. The comparison between the two strains of mice at baseline, including the summation of 30 min of travelled distance and the total time spent in the central area of the apparatus, can be found in Appendix A. 

The locomotor activity after ketamine injection was observed for 60 min. The travelled distance (cm) of each group is shown at the corresponding time points, by 2 min time intervals, in Figure 2A,B and Figure 3A,B. The total distance travelled during 60 min after ketamine injection is demonstrated in Figure 2C,D and Figure 3C,D. ANOVA tests and post hoc analysis were used to assess the differences of locomotor responses among each group. Table 1 summarizes the effect of different ketamine dosages on the total distance travelled during a 60 min period after ketamine injection.

### 3.1. Age Effect

ANOVA data showed a significant difference in the amount of distance traveled after receiving a ketamine injection between adolescent and adult C57BL/6G mice [F (1, 64) = 7.786, *p* < 0.01], but not in BALB/c [F (1, 62) = 0.057, *p* = 0.81]. In C57BL/6J strain mice treated with a low dose of ketamine (25 mg/kg), adolescents exhibited a more intense response of locomotor hyperactivity than adults. The summation of total travelled distance (in meters) following ketamine injection appeared to have a 3-fold increase in the adolescent (175.0 ± 13.8) and 2-fold increase in the adult (130.9 ± 11.8) group compared with the vehicle group (65.2 ± 7.0) (*p* < 0.001) according to Bonferroni’s post hoc test. When treated with a higher dose of ketamine (50 mg/kg), delayed responses in moving behaviors were detected. Similarly, adult C57BL/6J mice (136.9 ± 8.1) showed less intensity in total distance traveled than adolescents (171.2 ± 15.7), as shown in Figure 2 C,D and Table 1. However, in BALB/c mice, locomotor activity transiently increased in adults with the 25 mg/kg ketamine injection. The total travelled distance was higher in adult mice with the 25 mg/kg dose than the adolescents, but there was no difference following the 50 mg/kg ketamine treatment (shown in Figure 3 and Table 1).

### 3.2. Ketamine Dose Effect

ANOVA data showed a significant difference in the amount of distance traveled among the three dosage groups after receiving a ketamine injection [F (2, 126) = 16.67, *p* < 0.0001]. In C57BL/6J mice following the 25 mg/kg injection, both adolescents and adults exhibited a strikingly faster increase in locomotor hyperactivity compared to those who received 50 mg/kg. In mice treated with a 50 mg/kg dose of ketamine, there was a notable delay in peak locomotor activation, which occurred 30 min after the injection. For BALB/c mice, the acute and transient locomotor activation was observed only in the adult group when treated with a low dose (25 mg/kg) of ketamine. Conversely, when adult BALB/c mice were treated with a high dose (50 mg/kg) of ketamine, they traveled a relatively shorter distance in 60 min compared to the low dose (25 mg/kg) of ketamine (*p* < 0.05). This was determined by Bonferroni’s post hoc test, as shown in Figure 4 and Table 1.

### 3.3. Strain Variation

There were notable differences in locomotor behaviors between the C57BL/6J and BALB/c strains, as shown in Figure 2 and Figure 3. ANOVA revealed a significant difference in total traveled distance [F (2, 126) = 32.17, *p* < 0.0001] and average traveled speed [F (1, 126) = 32.16, *p* < 0.0001] between the two strains following ketamine injection, as shown in Figure 4. In adolescent mice, the average travelled speed over 60 min after ketamine treatment was about 2-fold higher in C57BL/6J compared to BALB/c mice (*p* < 0.001), according to Bonferroni’s post hoc test. For adult mice, there was no significant difference in average travelled speed between two strains after receiving a low dose (25 mg/kg) of ketamine. However, when given a high dose (50 mg/kg) of ketamine, adult BALB/c mice showed less travelled speed compared to adult C57BL/6J (*p* < 0.001).

## 4. Discussion

This integrative study aims to clarify the correlations of strain and age effects of ketamine by injecting low and high ketamine doses. The principal findings are as follows: (1) ketamine increased locomotor activity; (2) the pharmacological effects vary among different strains of mice, with hyperlocomotion consistently potentiated in the C57BL/6J group rather than in the BALB/c group; (3) adolescents demonstrated hypersensitive effects to ketamine in C57BL/6J mice; (4) both strains exhibited trends of dose-shift neurobehavioral changes, with acute onset following the low dose ketamine challenge in C57BL/6J mice.

### 4.1. Ketamine Induced Locomotor Change

The open field test is commonly used in rodent studies to evaluate anxiety and many neurobehaviors. In humans, ketamine can induce schizophrenic-like symptoms [37,38], and in animal studies, ketamine-induced locomotor hyperactivity may indicate excitatory behaviors and the psychotic domain for schizophrenia [35]. The plausible mechanism for how ketamine work is that it activates dopaminergic neurons via disinhibition of glutamatergic projections onto dopamine neurons. 

Studies have also shown that acute or chronic ketamine use increases locomotor activity with rising dopamine levels in the cortex, striatum, and nucleus accumbens [16]. Additionally, this locomotor activation may also indicate drug reward [35] through the activation of dopaminergic circuits that contribute to the reinforcing effects. Past research has used the effects of ketamine on locomotion as a means of assessing acute drug sensitivity, as well as chronic exposure-related neuroplasticity and sensitization in addiction models [39,40,41].

### 4.2. Interactions between Strain, Age, and Dosage Effects

In open field tests, various strains of mice have exhibited distinct anxiety and locomotor responses [42,43]. For instance, BALB/c mice have been observed to display less exploratory behavior than C57BL/6J mice [43]. Our study aligns with this discovery, as C57BL6/J mice spent a greater amount of time in the central area during the 30-min habituation period in comparison to BALB/c mice (as evidenced in Appendix A). 

Age differences also play an important role in our study. When treated with the same dosage of ketamine, adolescent C57BL/6J mice showed a higher peak response and longer duration of locomotor activation than adult C57BL/6J mice (Figure 2A,B). The literature suggests that a sub-anesthetic dosage of ketamine exceeding 10 mg/kg [28,35,44] could lead to an increase in ambulatory activities, particularly in adolescents when compared to adult animals [40,41,45]. Our findings are consistent with these reports, showing that adolescent mice are more sensitive to ketamine-related locomotor hyperactivity than adult mice. It is interesting to note that the locomotion of adult BALB/c mice was relatively low when treated with high dose (50 mg/kg) ketamine compared to the vehicle group. It is unlikely that this can be explained by the anesthetic effect of ketamine, which has been demonstrated to generally occur at doses above 100 mg/kg, resulting in initial immobilization followed by postanesthetic locomotion [32,46]. Instead, contributing factors may include the innate low sensitivity of BALB/c mice to ketamine-induced hyperlocomotion [43] and age difference. Many animal studies have demonstrated that various outcomes of ketamine seem to intensify during adolescence, such as emergent reactions of ketamine anesthesia, neurotoxic effects of high doses treatment, and antidepressant effects [41]. This study addresses the intriguing issue of discriminating age effects in the clinical application of ketamine for the treatment of refractory depression. 

Different dosages of ketamine were found to produce different time course effects of locomotor activation in previous studies [32,46]. In our study of C57BL/6J mice, a lower dose (25 mg/kg) of ketamine produced the rapid onset of locomotor activation, while a higher dose (50 mg/kg) demonstrated a delayed effect of hyperlocomotion. The initial suppression effect may be due to high dose ketamine producing transient anesthetic effects, but not in BALB/c mice. These findings are consistent with previous reports [32,46]. The rapid pharmacological response not only caters to psychological desire but may also be prone to the development of ketamine addiction [47].

### 4.3. Adolescent Vulnerability to Ketamine-Related Psychotomimetic Effects

During adolescence, the brain is in a critical stage of development, making individuals more vulnerable to drug addiction [39]. The vulnerable brain in this stage and lack of self-control can lead to sensation-seeking and risky behaviors, which in turn can drive substance abuse. Furthermore, studies show that repeated use of ketamine produces sensitizing effects in rats, increasing the likelihood of addiction [40,41]. In view of psychotomimetic effects, the C57BL/6J mice in our study showed hypersensitivity to hyperlocomotion, especially adolescents. Ketamine is commonly used as the usual model of schizophrenia [18,19], a complex brain disorder that involves gene–gene and gene–environmental interactions. Substance exposure during adolescence has been proposed to deteriorate brain differentiation, remodeling, pruning, and maturation through disturbing dopaminergic projection, glutamate moderation, and myelination [22]. As such, exposure to ketamine during this period may increase the risk of developing schizophrenia or other psychotic disorders. 

Some studies have demonstrated pharmacokinetic differences in drug biodistribution and metabolism depending on age. In particular, research suggests that adolescents have a shorter plasma half-life of many substances and psychotropics, both in human and in animal studies [40,48]. Furthermore, it has been observed that ketamine metabolism is slower in older rats than in adults [49]. These pharmacokinetic differences may influence the ketamine effects of anesthesia, pain reduction, and antidepressant effects [41,50,51,52]. Animal research has revealed changes in the densities of NMDA receptors over the lifespan. Specifically, the densities of NMDA receptors were found to increase from childhood to adolescence and then attenuate in certain brain areas during adulthood [53,54,55,56]. In this study, ketamine-induced hyperlocomotion may have originated from the interaction between ketamine and the NMDA system. A possibility is that the greater amount of NMDA receptors in the developmental shift predisposes the locomotor hypersensitivity of C57BL/6J adolescent mice [41]. 

The BALB/c strain has been observed to display anxious and aggressive behavior and differences in brain anatomy, such as hippocampal lamination defects and neuronal migration [57]. In this study, two strains of mice were used to symbolize how diverse human species may respond differently to ketamine exposure. Previous studies have also shown evidence of genetic factors in the differences of NMDA blockade responses between the two strains [28,58]. Recently, an altered endogenous tone of NMDA receptor-mediated neurotransmission was found in BALB/c mice, which was associated with a sociability deficit [27,59,60,61]. Pharmacodynamic diversity may be a factor to explain the discrepancy between the two mouse strains. Additionally, exploring the modulation of glutamatergic neurotransmission in BALB/c mice would be intriguing for psychiatry and neuropharmacological research.

### 4.4. Clinical Implications of the Study

Ketamine has been proven to exhibit a rapid antidepressant response and reduce suicidal ideation. However, the underlying neurobiological mechanism remains unclear, although numerous studies provide evidence that its treatment effects are achieved through modulating glutamate neurotransmission. The glutamate pathway also plays a role in memory consolidation by activating NMDA receptors and enhancing LTP formation. It also has diverse neuropathogenesis to modulate the development of neurocognitive disorders and provides novel opportunities for new drug discovery [54,55,56,62]. Our study demonstrates that adolescents may exhibit hypersensitivity to neurobehavioral changes, indicating vulnerability to memory formation, addiction tendencies, and psychotomimetic effects. Furthermore, experimental results reveal diverse pharmacological responses to ketamine in different strains, suggesting potential pharmacological variations in different populations. Based on these research findings, it is important to note the potential negative impacts of ketamine on learning, memory formation, and addiction in adolescents. Therefore, future research on ketamine or ketamine-like agents should consider genetic background and developmental stage differences to enable precise medicine approaches.

### 4.5. Limitation

The limitations of this study are as follows: (1) Acute injection could not reflect the real situation of ketamine abuse behaviors. (2) There was a lack of chronic ketamine effects to explore behavioral sensitization and neuroadaptation. (3) The open field test could only imply ketamine-related locomotion change; it will be important to investigate the other symptom domains by using more neurobehavioral models in the future. (4) If possible, it would be beneficial to have access to neuroimaging or neurochemistry data, as this could provide further insights into the mechanisms underlying neurobehavior. (5) Ketamine is a mixture of (R)-ketamine and (S)-ketamine, which have different affinities to the NMDA receptor and produce distinct neuropsychopharmacological effects. It would be helpful to clarify the correlations between strain and age effects on ketamine isomers.

## 5. Conclusions

The results of our study demonstrate that ketamine-induced locomotor behavior is influenced by both dose and age. The observed differences in neurobehavioral responses between the two mouse strains may suggest that sensitivity to ketamine is biologically determined. Furthermore, individual vulnerability to ketamine may indicate diverse pharmacological responses in different populations. It is noteworthy to consider the potential negative consequences of ketamine on learning, memory formation, addiction, and susceptibility to the development of psychosis, particularly in adolescents who are in a critical stage of brain development.

## Figures and Tables

**Figure 1 biomedicines-11-01821-f001:**
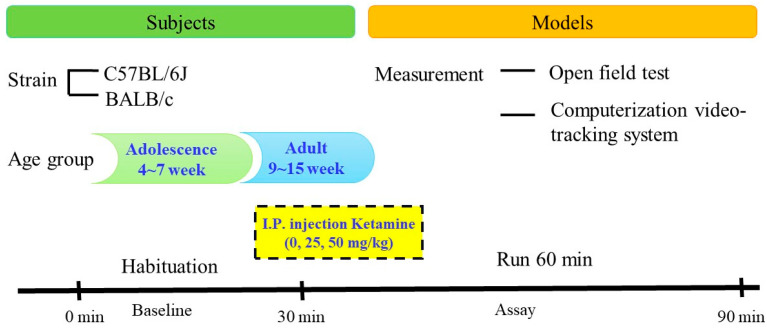
The study procedure: After 30 min baseline habituation, mice were followed by intraperitoneal injection with different ketamine dosages. The mice then were placed in the perimeter and allowed to explore the apparatus for 60 min. A video-tracking system was adapted to measure the motor response in the open field.

**Figure 2 biomedicines-11-01821-f002:**
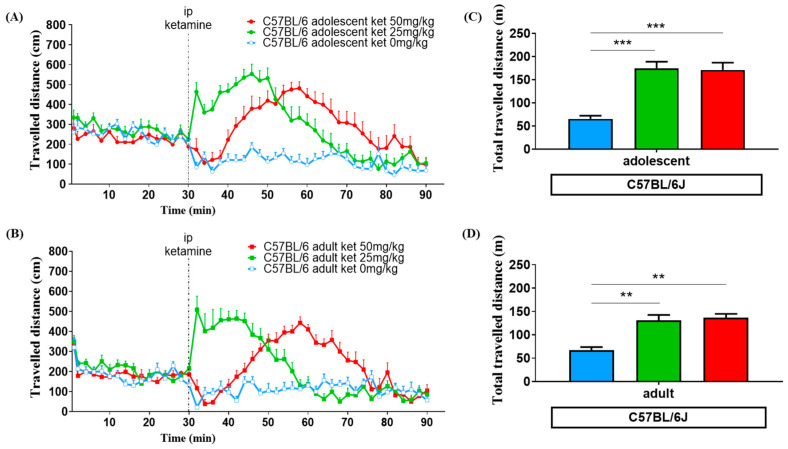
Travelled distance (cm) at the corresponding time points in the open field test for adolescent (**A**) and adult (**B**) C57BL/6J mice treated with different ketamine dose. The groups are denoted as adolescent (circle) and as adult (square), as well as different ketamine doses: 0 mg/kg (blue), 25 mg/kg (green), 50 mg/kg (red). Total travelled distance (m) represents the summation of moving distance following ketamine injection running for 60 min in both age groups: (**C**) for adolescent, and (**D**) for adulthood. Vertical bars represent standard error of the mean (SEM). Differences among the study groups were evaluated by analysis of variance (ANOVA) and post hoc tests. Statistically significant differences between groups: ** *p* < 0.01, *** *p* < 0.001 vs. vehicle.

**Figure 3 biomedicines-11-01821-f003:**
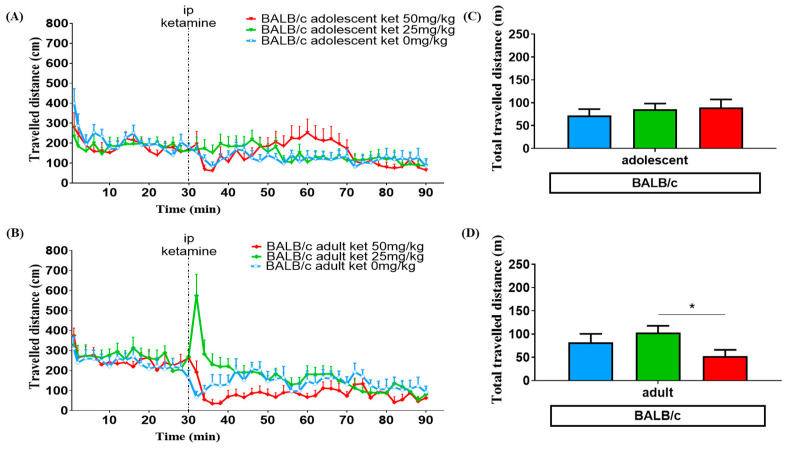
Travelled distance (cm) at the corresponding time points in the open field test for adolescent (**A**) and adult (**B**) BALB/c mice treated with different ketamine doses. The groups are denoted as adolescent (inverted triangle) and as adult (prism), as well as different ketamine doses: 0 mg/kg (blue), 25 mg/kg (green), 50 mg/kg (red). Total travelled distance (m) represents the summation of moving distance following ketamine injection running for 60 min in both age groups: (**C**) for adolescent, and (**D**) for adulthood. Vertical bars represent SEM. Differences among the study groups were evaluated by ANOVA and post hoc tests. Statistically significant differences between 25 mg and 50 mg dose groups: * *p* < 0.05.

**Figure 4 biomedicines-11-01821-f004:**
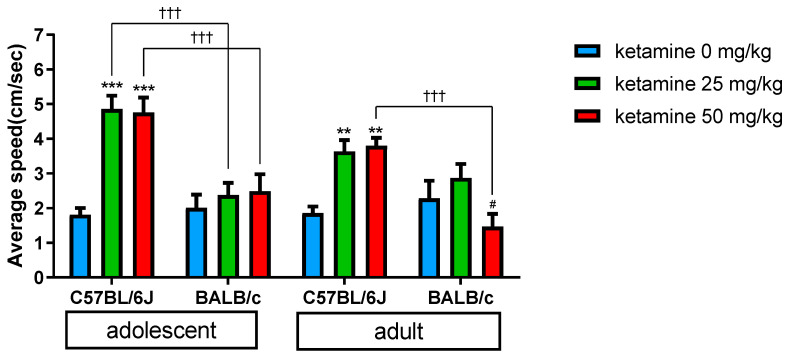
Comparisons of the average travelled speed (cm/sec) between C57BL/6J and BALB/c mice treated with 0, 25, and 50 mg/kg of ketamine. The differences of locomotor activity were measured following the ketamine injection running for 60 min in the open field. Vertical bars represent SEM. Differences among the study groups were evaluated by ANOVA and post hoc tests. Statistically significant differences between groups: ^#^
*p* < 0.05 vs. BALB/c treated with 25 mg/kg ketamine, ** *p* < 0.01, *** *p* < 0.001 vs. vehicle; ^†††^
*p* < 0.001 vs. BALB/c.

**Table 1 biomedicines-11-01821-t001:** Comparisons of locomotor activity between C57BL/6J and BALB/c mice treated with different ketamine doses.

Ketamine	C57BL/6J		BALB/c	
	Adolescent	Adult	Adolescent	Adult
0 mg/kg	65.2 ± 7.0	66.8 ± 6.8	72.3 ± 13.8	82.3 ± 18.2
25 mg/kg	175.0 ± 13.8 ^a^	130.9 ± 11.8 ^b^	85.5 ± 12.8	103.4 ± 14.5
50 mg/kg	171.2 ± 15.7 ^a^	136.9 ± 8.1 ^b^	89.6 ± 17.5	52.8 ± 13.3 ^c^

Values are means ± standard error of the mean (SEM); data represent the total travelled distance (meter) in 60 min following the ketamine injection. Differences among the study group were evaluated by multiple analysis of variance and post hoc tests. Subject numbers were 10–13 in each group. ^a^
*p* < 0.001 compared with 0 mg/kg ketamine group in adolescent C57BL/6J. ^b^
*p* < 0.01 compared with 0 mg/kg ketamine group in adult C57BL/6J, ^c^
*p* < 0.05 compared with 25 mg/kg ketamine group in adult BALB/c.

## Data Availability

Data are available upon request to the corresponding author.

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
