# Peer review of "Age, Dose, and Locomotion: Decoding Vulnerability to Ketamine in C57BL/6J and BALB/c Mice"

_biomedicines, 2023, doi:10.3390/biomedicines11071821_

Round 1

Reviewer 1 Report

Chen and colleagues in the present article entitled ‘Vulnerability to Ketamine-induced Locomotor Hyperactivity between C57BL/6J and BALB/c: Modulation by Age and Dose', investigated the neurobehavioral changes following ketamine injection in mice with different age and strains. Results showed that following ketamine injection, the travelled distance and speed significantly increased in C57BL/6J mice between both age groups (p < 0.01), but did not show in BALB/c. The pattern of hyperlocomotion showed delayed response in higher dose (50 mg/kg) compared to lower dose (25 mg/kg) ketamine treatment. Ketamine accentuated the locomotor activation in the adolescent C57BL/6J mice compared to the adult, but not in BALB/c strain.

In general, I think the idea of this article is really interesting and the authors’ fascinating observations on this timely topic may be of interest to the readers of Biomedicines. However, some comments, as well as some crucial evidence that should be included to support the author’s argumentation, needed to be addressed to improve the quality of the manuscript, its adequacy, and its readability prior to the publication in the present form. My overall judgment is to publish this paper after the authors have carefully considered my suggestions below, in particular reshaping parts of the ‘Introduction’ and ‘Methods’ sections by adding more evidence.

Please consider the following comments:

I suggest changing the title. In my opinion, in the present form it is too wordy and it seems to be not enough clear and specific. 

Abstract: According to the Journal’s guidelines, the abstract should be a total of about 200 words maximum and should be presented as a single paragraph, without sub-headings. Please correct the actual one. Also, in my opinion, Authors should consider rephrasing this section. According to the Journal’s guidelines, the Abstract should contain most of the following kinds of information in brief form. Please, consider giving a more synthetic overview of the paper's key points: I would suggest rephrasing the results and conclusion to make them clear for readers to understand.

A graphical abstract that will visually summarize the main findings of the manuscript is highly recommended.

Please provide the full names before using abbreviations.

In general, I recommend authors to use more references to back their claims, especially in the Introduction of this research article, which I believe is lacking. Thus, I recommend the authors to attempt to expand the topic of their article, as the bibliography is too concise. Nevertheless, I believe that less than 60/70 articles are too low for a research article. Therefore, I suggest the authors to focus their efforts on researching relevant literature: in my opinion, adding more citations will help to provide better and more accurate background to this study. 

Introduction: The ‘Introduction’ section is well-written and nicely presented, with a good balance of descriptive text and information about ketamine’s mechanisms of action and how it impacts on cognitive functions. Nevertheless, I believe that more information about ketamine’s neurobiological and biochemical underpinnings mediating its antidepressant effects will provide a better and more accurate background, because as it stands, this information is not highlighted in the text. I suggest to begin with a theoretical explanation of pharmacokinetics and pharmacodynamics of ketamine and of its mechanisms of action in specific brain areas, like prefrontal cortex. In this regard, I would suggest to add more information on changes in neuronal activity patterns in the cerebral cortex of animal and human models after ketamine administration (https://doi.org/10.3390/biomedicines10123189). In my opinion, authors could further explore relationship between the molecular regulation of higher-order neural circuits and neuropathological alterations that may be linked to ketamine effects (https://doi.org/10.3390/biomedicines10081897), in order to provide more insights on ketamine-induced activity switch in key brain regions tied to depression that may impact our understanding of ketamine’s treatment effects and future research in the field of neuropsychiatry.

Results: I suggest rewriting this section more accurately. To properly present experimental findings, I think that authors should provide full statistical details (like degree of freedom or post-hoc utilized), to ensure in-depth understanding and replicability of the findings.

Discussion: In this final section, authors described the results of their study and their argumentation and captured the state of the art well; however, I would have liked to see some views on a way forward. I believe that the authors should make an effort, trying to explain the theoretical implication as well as the translational application of this paper, to adequately convey what they believe is the take-home message of their study. In this regard, I believe that it would be necessary to discuss theoretical and methodological avenues in need of refinement, as well as suggestions of a path forward in understanding of the effects of ketamine agents on locomotive changes.

In my opinion, I think the ‘Conclusions’ paragraph would benefit from some thoughtful as well as in-depth considerations by the authors, because as it stands, it is very descriptive but not enough theoretical as a discussion should be. Authors should make an effort, trying to explain the theoretical implication as well as the translational application of their research.

Figures: I suggest to modify all figures for clarity because, as it stands, the readers may have difficulty comprehending it. Also, please change the scale of the vertical axis and use the same minimum/maximum scale value in all the graphs.

References: Authors should consider revising the bibliography, as there are several incorrect citations. Indeed, according to the Journal’s guidelines, they should provide the abbreviated journal name in italics, the year of publication in bold, the volume number in italics for all the references. 

I hope that, after these careful revisions, this paper can meet the Journal’s high standards for publication. 

I am available for a new round of revision of this article. 

Best regards,

Reviewer

Author Response

Reviewer 1#

General Comments for Authors

Chen and colleagues in the present article entitled ‘Vulnerability to Ketamine-induced Locomotor Hyperactivity between C57BL/6J and BALB/c: Modulation by Age and Dose', investigated the neurobehavioral changes following ketamine injection in mice with different age and strains. Results showed that following ketamine injection, the travelled distance and speed significantly increased in C57BL/6J mice between both age groups (p < 0.01), but did not show in BALB/c. The pattern of hyperlocomotion showed delayed response in higher dose (50 mg/kg) compared to lower dose (25 mg/kg) ketamine treatment. Ketamine accentuated the locomotor activation in the adolescent C57BL/6J mice compared to the adult, but not in BALB/c strain.

In general, I think the idea of this article is really interesting and the authors’ fascinating observations on this timely topic may be of interest to the readers of Biomedicines. However, some comments, as well as some crucial evidence that should be included to support the author’s argumentation, needed to be addressed to improve the quality of the manuscript, its adequacy, and its readability prior to the publication in the present form. My overall judgment is to publish this paper after the authors have carefully considered my suggestions below, in particular reshaping parts of the ‘Introduction’ and ‘Methods’ sections by adding more evidence.

We are grateful for Reviewer’s constructive suggestions. We rephrased some sentences and highlighted the study aims and evidence in the whole manuscript.

Response to the specific comments:

Q #1. I suggest changing the title. In my opinion, in the present form it is too wordy and it seems to be not enough clear and specific.

As suggestion, we changed the title to “Vulnerability to Ketamine-induced Hyperlocomotion between C57BL/6J and BALB/c: Modulation by Age and Dose”

Q #2. Abstract: According to the Journal’s guidelines, the abstract should be a total of about 200 words maximum and should be presented as a single paragraph, without sub-headings. Please correct the actual one. Also, in my opinion, Authors should consider rephrasing this section. According to the Journal’s guidelines, the Abstract should contain most of the following kinds of information in brief form. Please, consider giving a more synthetic overview of the paper's key points: I would suggest rephrasing the results and conclusion to make them clear for readers to understand.

For the concise presentation, the abstract was rephased and rewritten as a single paragraph without subheadings in the revised manuscript. The New abstract is 197 words according to the guidelines of the journal.

Q #3. A graphical abstract that will visually summarize the main findings of the manuscript is highly recommended.

The study main finding is shown in the revised manuscript as following:

In conclusion, ketamine induced locomotor behavior is modulated by dose and age. The discrepancy of neurobehaviors in the two strain mice indicates sensitivity to ketamine is biologically determined. This study suggests individual vulnerability to ketamine’s pharmacological responses varies biologically.

Q #4. Please provide the full names before using abbreviations.

        We have reviewed and rechecked thorough manuscript and provided the full name before using the abbreviations.

Q#5. In general, I recommend authors to use more references to back their claims, especially in the Introduction of this research article, which I believe is lacking. Thus, I recommend the authors to attempt to expand the topic of their article, as the bibliography is too concise. Nevertheless, I believe that less than 60/70 articles are too low for a research article. Therefore, I suggest the authors to focus their efforts on researching relevant literature: in my opinion, adding more citations will help to provide better and more accurate background to this study.

In light of recent research on the neurobiological and biochemical mechanisms associated with ketamine, our revised introduction now includes pertinent information supporting the notion of shared biological underpinnings between depression and addiction. Given the increasing attention that ketamine has garnered as a treatment for refractory depression across diverse populations and developmental stages, it is crucial to thoroughly weigh the risks and benefits. Relevant references have been cited as Ref. 9-12, 19 and 24-26 to substantiate our findings.

Q #6. Introduction: The ‘Introduction’ section is well-written and nicely presented, with a good balance of descriptive text and information about ketamine’s mechanisms of action and how it impacts on cognitive functions. Nevertheless, I believe that more information about ketamine’s neurobiological and biochemical underpinnings mediating its antidepressant effects will provide a better and more accurate background, because as it stands, this information is not highlighted in the text. I suggest to begin with a theoretical explanation of pharmacokinetics and pharmacodynamics of ketamine and of its mechanisms of action in specific brain areas, like prefrontal cortex. In this regard, I would suggest to add more information on changes in neuronal activity patterns in the cerebral cortex of animal and human models after ketamineadministration (https://doi.org/10.3390/biomedicines10123189). In my opinion, authors could further explore relationship between the molecular regulation of higher-order neural circuits and neuropathological alterations that may be linked to ketamine effects (https://doi.org/10.3390/biomedicines10081897), in order to provide more insights on ketamine-induced activity switch in key brain regions tied to depression that may impact our understanding of ketamine’s treatment effects and future research in the field of neuropsychiatry.

We appreciate the valuable suggestions provided by the Reviewer regarding the clinical implications of ketamine for depression treatment and its underlying pharmacological mechanisms. In the revised manuscript, we have emphasized the issue and thoroughly addressed the pharmacokinetics and pharmacodynamics of ketamine. We have also incorporated relevant references to support our findings and have taken into consideration the diverse pharmacological responses observed in different populations. Additionally, we have highlighted the overlapping neural circuits involved in depression and addiction. It is important to note the potential risks and benefits of ketamine treatment in various depression subgroups (shown in Introduction, second paragraph, Line 49-68).

Q #7. Results: I suggest rewriting this section more accurately. To properly present experimental findings, I think that authors should provide full statistical details (like degree of freedom or post-hoc utilized), to ensure in-depth understanding and replicability of the findings.

To accurately convey the experimental findings, we made efforts to rephrase and rewrite the results section. Additionally, the data underwent reanalysis using analysis of variance (ANOVA) followed by Bonferroni's post hoc tests. The revised manuscript includes comprehensive statistical values and information. Furthermore, as per the suggestion of the Academic Editor, the figures have been replaced with color format  (shown in the Results section, Figures 1-4, Lines 152-242).

Q #8. Discussion: In this final section, authors described the results of their study and their argumentation and captured the state of the art well; however, I would have liked to see some views on a way forward. I believe that the authors should make an effort, trying to explain the theoretical implication as well as the translational application of this paper, to adequately convey what they believe is the take-home message of their study. In this regard, I believe that it would be necessary to discuss theoretical and methodological avenues in need of refinement, as well as suggestions of a path forward in understanding of the effects of ketamine agents on locomotive changes.

Based on this research findings, the last paragraph of discussion regarding the clinical implication was rewrote and highlighted the issue of negative impacts in adolescent and subject variations to ketamine pharmacological effect. As following: 

Ketamine has been proven to exhibit a rapid antidepressant response and reduce suicidal ideation. However, the underlying neurobiological mechanism remains unclear, although numerous studies provide evidence that its treatment effects are achieved through modulating glutamate neurotransmission. The glutamate pathway also plays a role in memory consolidation by activating NMDA receptors and enhancing LTP formation. Apart from its antidepressant effects, our study demonstrates that adolescents may exhibit hypersensitivity to neurobehavioral changes, indicating vulnerability to memory formation, addiction tendency, and psychotomimetic effects. Furthermore, experimental results reveal diverse pharmacological responses to ketamine in different strains, suggesting potential pharmacological variations in different populations. Based on these research findings, it is important to note the potential negative impacts of ketamine on learning, memory formation, and addiction in adolescents. Therefore, future research on ketamine or ketamine-like agents should consider genetic background and developmental stage differences to enable precise medicine approaches (shown in Line 336-350).

Q #9. In my opinion, I think the ‘Conclusions’ paragraph would benefit from some thoughtful as well as in-depth considerations by the authors, because as it stands, it is very descriptive but not enough theoretical as a discussion should be. Authors should make an effort, trying to explain the theoretical implication as well as the translational application of their research.

As Reviewer’s suggestion, we tried to extend the implication and new insight for the individual variation and potential negative impacts of ketamine based on our study results. As following:

Furthermore, individual vulnerability to ketamine may indicate diverse pharmacological responses in different populations. It is noteworthy to consider the potential negative consequences of ketamine on learning, memory formation, addiction, and susceptibility to the development of psychosis, particularly in adolescents who are in a critical stage of brain development (shown in Conclusion, Line 365-369).

Q #9. Figures: I suggest to modify all figures for clarity because, as it stands, the readers may have difficulty comprehending it. Also, please change the scale of the vertical axis and use the same minimum/maximum scale value in all the graphs.

To avoid confusion when comparing between multiple groups in Figure 2, 3 and 4, the figure has been changed to colored. Furthermore, the maximum scale of the vertical axis has been checked to ensure that the same minimum/maximum scale values are used for the same comparison units.

Q #10. References: Authors should consider revising the bibliography, as there are several incorrect citations. Indeed, according to the Journal’s guidelines, they should provide the abbreviated journal name in italics, the year of publication in bold, the volume number in italics for all the references. 

All the references were rechecked and edited to fit the Journal bibliographic format, such as for the year of publication (in bold), the abbreviated journal name (in italics), and the volume number (in italics).

Reviewer 2 Report

The MS presented tackles an important issue: the characterization of the behavioral changes caused by ketamine on different mice strains.

While this is an important subject, I consider there are serious issues with the paper that I will list below.

- I consider the most important shortcoming to be the very limited conclusions that are provided by the experiments carried out. While the MS constantly reminds us of the many different positive and negative effects of ketamine, in reality there is only one protocol that tests locomotion, which provides very limited information to yield significant conclusions.

- the comparison of locomotion between groups is severely compromised because two different arenas are used (for example, white spaces tend to be more anxiogenic than darker spaces)

- The presentation of the results is done through 3 figures and 1 table, but they are mostly redundant, showing the same data in different styles. Figures 2 and 3 would be enough to show the data.

- The conclusion overstates the conclusions that can be taken from the study, trying to draw conclusions about anxiety (l.223), depression (l.254), and addcition (l.264), for example.

I honestly think that this study could be part of a very interesting and important characterization of different behavioral effects of different doses of ketamine in different mouse strains, including anxiety, locomotion, and even anti-depressive-like effects. Those tests can be done on the same animals and would greatly enrich the conclusions drawn, living up to the statement "clarify the correlations of strain and age effects of ketamine" (l.214).

- On a side note, the authors refer that open field test can be used to assess anxiety (l.223), but not by using total locomotion. I recommend going back to the original video recordings and measure the time and distance spent in the center area of the open field: that would be a primary measurement of anxiety.

 A few more suggestions:

- the first paragraph of Results includes mostly protocol description

- further discuss the reasons behind the delayed increase in locomotion in the 25 mg/kg C57 group

- the pos hoc test choice seems odd as it is rarely used in the field and only pairwise comparisons seem necessary (Bonferroni and Tukey seem more appropriate)

- l.250; 50 mg/kg ketamine actually decreases locomotion, so it should not be justified with a decrease in sensitivity to hyperlocomotor effects of ketamine  

- The text needs extensive English review. While understandable from beginning to end, it requires that the reader tries to guess the meaning behind the words in many phrases, and undermines the precision of scientific writing.

- The references should be more up-to-date. The most recent papers cited are from 2018, and most from before 2015. Much research (probably most of research) on ketamine has been carried since then.

I found particularly interesting the discussion on NMDA in adolescence to explain the differences observed (l.284), and Balb/C neurophisiology(l.294).

Author Response

Reviewer 2#

General Comments for Authors

The MS presented tackles an important issue: the characterization of the behavioral changes caused by ketamine on different mice strains. While this is an important subject, I consider there are serious issues with the paper that I will list below.

Q #1. I consider the most important shortcoming to be the very limited conclusions that are provided by the experiments carried out. While the MS constantly reminds us of the many different positive and negative effects of ketamine, in reality there is only one protocol that tests locomotion, which provides very limited information to yield significant conclusions.

      The study was designed to assess the locomotor behaviors between C57BL/6J and BALB/c treated with different ketamine doses. Based on our previous experiences, the open-field test is a powerful experiment to measure the drug sensitivity. Under the well systemic, negative and positive controls, the mouse movement was continuously measured by the computerized video-tracking system and automatically generated the parameters by computer programs. Therefore, we could make sure the reliable and valid results to assess the ketamine sensitivity between different strain and age.

Thank you for reminding us to always keep multidimensional drug-related behavioral change in mind. We acknowledge that our study has limitations and have included this point in the "Limitations" section. However, we also hope that our research framework can contribute to the fundamental idea of precise medicine in ketamine research. Our study design shows that the different effects of ketamine are biologically determined, as frequently observed in clinical situations. We are excited to see more neurobehavioral animal models being used in neurocircuitry and neurobiology studies of ketamine with different biological entities. This will help address the benefits and drawbacks with more accurate mechanistic verification.

Q #2. the comparison of locomotion between groups is severely compromised because two different arenas are used (for example, white spaces tend to be more anxiogenic than darker spaces)

It is questionable issue for potential bias between the white and black arenas as the Reviewer pointed out. The study aims are to compare the ketamine-induced behavioral reactions between age and two mouse strains. The experiments were designed for 30-minute habituation as internal and systemic controls.

To address the reviewer's concerns, we conducted a reanalysis of the parameters between the strains in the baseline. Specifically, we reviewed the original video records and analyzed the baseline locomotor activity, paying particular attention to the duration of time that the mice spent in the central area. We found that C57BL6/J mice spent a significantly greater amount of time in the central area of the white box compared to BALB/c mice in the dark box, which contradicts the intuitive concept of anxiogenic effects in white spaces for C57BL6/J mice. These results suggest the presence of innate biological variations between the two strains (as depicted in Supplemental Figure 2).     

Q #3. The presentation of the results is done through 3 figures and 1 table, but they are mostly redundant, showing the same data in different styles. Figures 2 and 3 would be enough to show the data.

We analyzed the distance travelled and time spent in the central area of the open field to confirm the pattern of locomotion during the habituation period. The results are presented in Supplements 1 and 2. As Reviewer suggestion, Figure 4 shares similarities with Figures 2 and 3 to enable a comparison between all groups at once. Upon consideration, we have modified Figure 4 from analyzed distance traveled to average speed after ketamine treatment.

Q #4. The conclusion overstates the conclusions that can be taken from the study, trying to draw conclusions about anxiety (l.223), depression (l.254), and addcition (l.264), for example.

Based on the experimental results, the conclusion was revised and highlighted the strain variations and age effect of ketamine sensitivity (shown in conclusion, L 363-370). The conclusion has been revised to avoid overestimating the impact of ketamine on neurodifferentiation. Future studies can address the specific impacts of ketamine with identified and hypothetical neurobiological mechanisms.

Q #5. I honestly think that this study could be part of a very interesting and important characterization of different behavioral effects of different doses of ketamine in different mouse strains, including anxiety, locomotion, and even anti-depressive-like effects. Those tests can be done on the same animals and would greatly enrich the conclusions drawn, living up to the statement "clarify the correlations of strain and age effects of ketamine" (l.214).

As Reviewer suggestions, it would be more comprehensive to clarify the neurobehavioral alterations through multidimensional experiments. In the revised manuscript, we put the issue in the study limitation section (shown in L 353-362).

Q #6. On a side note, the authors refer that open field test can be used to assess anxiety (l.223), but not by using total locomotion. I recommend going back to the original video recordings and measure the time and distance spent in the center area of the open field: that would be a primary

It is insightful that the reviewer reminded us that different backgrounds in the open field may elicit varying anxiety responses in mice. Additionally, the open field itself provides an opportunity to verify the influence of possible systemic bias. We went back to the original video records and analyzed the baseline locomotor activity, especially the duration mice spent in the central area. In comparing the central duration in the open field, C57BL6/J mice in the possible anxiogenic-white box significantly spent more time in the central area compared to BALB/c mice in the dark box (shown in supplements). This finding was presented as Supplement 1 and may indicate that the possible anxiogenic effect of the white box is not a significant confounding factor in our test when interact with the different mice strain.

 A few more suggestions:

  1. the first paragraph of Results includes mostly protocol description

In this study, we begin the results section by outlining the number of groups formed during the grouping process, given the many factors involved. Following this, in the section about habituation, we have addressed the reviewer's feedback by comparing baselines of two different strain mice, rather than providing an excessive description of the protocol. We have also taken into account the reviewer's reminder that different colored boxes may elicit different stimuli in the mice, and have reviewed video footage of mice entering the central area to assess the potential effects of differently colored apparatus on their anxiety levels. Our findings on this matter are presented in the supplementary material. In the following results section, we focus on comparing anxiety levels after ketamine administration.

  1. further discuss the reasons behind the delayed increase in locomotion in the 25 mg/kg C57 group

The higher dose of ketamine would produce the transient anesthetic effect and reversed 10 minutes later. We added the point in discuss (shown in revised manuscript, L293-295).

  1. the pos hoc test choice seems odd as it is rarely used in the field and only pairwise comparisons seem necessary (Bonferroni and Tukey seem more appropriate)

To accurately convey the experimental findings, we made efforts to rephrase and rewrite the results section. Additionally, the data underwent reanalysis using analysis of variance (ANOVA) followed by Bonferroni's post hoc tests. The revised manuscript includes comprehensive statistical values and information.

  1. 250; 50 mg/kg ketamine actually decreases locomotion, so it should not be justified with a decrease in sensitivity to hyperlocomotor effects of ketamine  

We appreciate the reviewer's suggestion that the results of high-dose keramine treatment may not accurately describe a reduction in sensitivity to hyperlocomotion. We have made efforts to avoid such an interpretation. The reviewer has highlighted that certain groups that received high doses exhibited decreased locomotion, and we have reexamined the data accordingly. For C57BL/6J mice, 50mg/kg ketamine treatment indeed showed hyperlocomotion. An even more impressive finding is the delayed response observed with the 50mg/kg treatment compared to the 25mg/kg treatment. This serves as a reminder that even with recreational use, we should be aware of the potential sensitivity to psychotic characteristics and the possibility that a rapid response may increase the risk of addiction. Regarding BALB/c mice treated with 50mg/kg in adulthood, there is a trend towards decreased locomotion, as the reviewer has emphasized, but no difference observed compared to the vehicle. However, the difference between the 50mg/kg and 25mg/kg treatments suggests that adult BALB/c mice may be more susceptible to higher doses of ketamine in certain pharmacological aspects.

  1. The text needs extensive English review. While understandable from beginning to end, it requires that the reader tries to guess the meaning behind the words in many phrases, and undermines the precision of scientific writing.

The manuscript has been edited using Notion A.I. to improve its scientific expression. We have made an effort to use appropriate terminology and avoid ambiguity. We appreciate the reviewer's reminder.

  1. The references should be more up-to-date. The most recent papers cited are from 2018, and most from before 2015. Much research (probably most of research) on ketamine has been carried since then.

As pointed out by the reviewer, research on ketamine has seen a significant increase in recent years. In this version of update, we aimed to enhance the current understanding of ketamine research, including an improved comprehension of ketamine's underlying mechanism and its impact on brain neural circuits. Furthermore, the other reviewer suggested exploring the potential of combining neurophysiology to identify biomarkers in depression research, which we have cited as references 25 and 26. Our update includes additional citations, such as Ref. 9-12, 19 and 24 we have included articles published as recently as 2022. We appreciate your reminder, as it has encouraged us to explore this research topic with a refreshed perspective.

  1. I found particularly interesting the discussion on NMDA in adolescence to explain the differences observed (l.284), and Balb/C neurophisiology(l.294).

The pharmacological effects of ketamine have led to research on the glutamate system in neuro-psycho pharmacology in recent decades. In the discussion section, we consider the impact of age and strain on the ketamine effect in relation to the NMDA system. This is based on the clinical observation that adolescents who frequently abuse ketamine often exhibit psychotomimetic effects, while NMDA hypofunction is currently the mainstream model of schizophrenia. In addition, ketamine's treatment of depression is also closely related to NMDA, and these two groups show clear age differences. Therefore, from a developmental perspective, we believe that investigating the maturity of the NMDA system is a promising direction for future studies on ketamine or similar drugs. Furthermore, our findings suggest that BALB/c mice may have innate differences in the NMDA system, which could be explored in future disease models or drug research.

Round 2

Reviewer 1 Report

The authors did an excellent job clarifying all the questions I have raised in my previous round of review. Currently, this paper entitled ‘Vulnerability to Ketamine-induced Hyperlocomotion between C57BL/6J and BALB/c: Modulation by Age and Dose’, is a well-written, timely piece of research that described neurobehavioral changes following ketamine injection in mice with different age and strains.

Overall, this is a timely and needed work. It is well researched and nicely written, therefore I believe that this paper does not need a further revision, therefore the manuscript meets the Journal’s high standards for publication.

I am always available for other reviews of such interesting and important articles.

Thank You for your work, Reviewer

Author Response

Comments and Suggestions for Authors

The authors did an excellent job clarifying all the questions I have raised in my previous round of review. Currently, this paper entitled ‘Vulnerability to Ketamine-induced Hyperlocomotion between C57BL/6J and BALB/c: Modulation by Age and Dose’, is a well-written, timely piece of research that described neurobehavioral changes following ketamine injection in mice with different age and strains.

Overall, this is a timely and needed work. It is well researched and nicely written, therefore I believe that this paper does not need a further revision, therefore the manuscript meets the Journal’s high standards for publication.

I am always available for other reviews of such interesting and important articles.

Thank You for your work. 

Response to REVIEWER: 

We sincerely appreciate your review and constructive feedback for our manuscript.